# Balancing Precision and Richness in Image Caption Services for Enhanced Descriptive Accuracy

## Abstract

Current image captioning services often learn to generate captions by imitating ground truth references, which are constrained by the limitations of manual annotations. This leads to overlooked details in images, causing captions to lack richness and precise descriptions, critical for enhanced image captioning services. To address this, we propose a CLIP-based image captioning framework designed to balance descriptive precision and richness enhancement. Our approach uses fine-grained pseudo tags for learning and integrates an asymmetric attention multimodal projector to map and fuse information across modalities effectively. We also introduce an evaluation metric, Tags Coverage, to measure the granularity of generated captions and incorporate it into reinforcement learning to optimize the reward function. This eliminates the need for additional text annotations while addressing unannotated details. Experimental results on the MS-COCO Karpathy's test set demonstrate the model's effectiveness, with improvement in CIDEr and Tags Coverage compared to state-of-the-art baselines, highlighting its potential for advancing precision and richness in image captioning services.

## 1 Introduction

The field of image captioning has advanced rapidly, emphasizing the integration of cross-modal feature knowledge to support service-driven applications. These services require robust visual perception, natural language generation, and effective multimodal feature alignment to deliver accurate and rich descriptions. Models such as (Vinyals et al., 2015; Ma et al., 2015) utilize CNNs to extract visual features and process them into textual captions. Approaches like Up-Down (Anderson et al., 2018) and Xmodal-Ctx (Kuo & Kira, 2022) enhance precision in descriptions through object detection and context integration. Despite achieving significant metric improvements, existing methods often rely on limited human annotations, leading to captions that lack descriptive richness and fail to balance precision and detail effectively (Rotstein et al., 2024). For real-world service contexts, such as news reporting or visual assistance, solutions must address these limitations by generating captions that incorporate nuanced image details. The focus on balancing precision and richness directly supports scalable service systems for diverse applications in the service computing.

Existing human annotations are often concise, typically describing images in a limited format that simplistically addresses "what is happening in where". For instance, as illustrated in Fig. 1, annotations usually only hit the main objects in the image, such as "*men*", "*horse*", "*crowd*", and "*costume*", yet miss other intricate contextual background details like the "*parade*", "*houses and windows behind the street*", or "*individuals walking in the front with flag*". From the aspect of human perceptions and cognition, rich and diverse contents would be described when human looks at the same image. This does not mean that longer or overly detailed descriptions are better, as too much complexity can lead to confusion. Instead, the captioner should make the generated sentences free from the rigid structure in a more rich and flexible mode. The current features extraction methods and training strategies may lead to perceptual and generative biases. After extracting visual information from the images, captioners often prioritize generating sentences closely imitating the ground truth. However, this approach tends to overlook the inherent details in images, excluding vocabularies that extend beyond the ground truth but accurately capture the image content. Consequently, this results in image captions that lack details, flexibility, and richness.

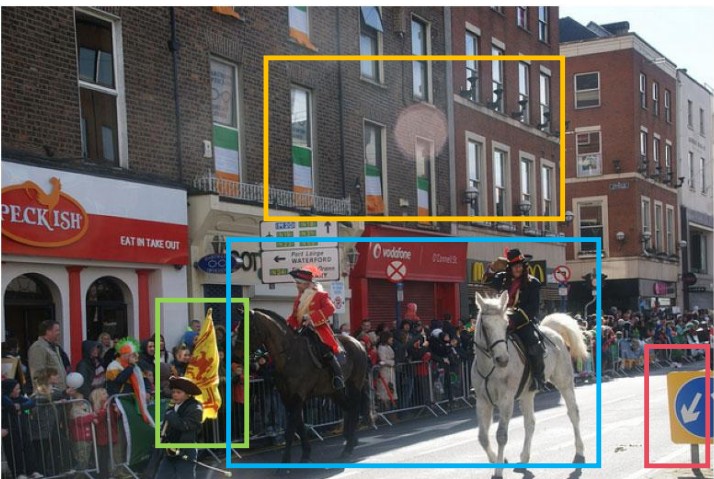

**Human annotation**

- A couple of men riding horses down a street with tall buildings.
- Men riding on horses in street next to buildings.
- A crowd is watching horses go down the street.
- A man dressed in red riding a horse through town.
- people in costume riding down the road on horses.

**Hit**                                    **Miss**

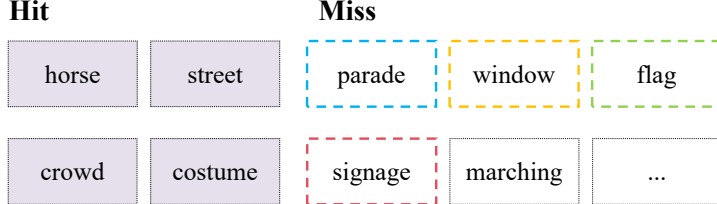

Figure 1: An example showcasing the limitations of human annotations - some vocabularies they successfully hit and what they miss reflecting in the corresponding color boxes in the image.

In order to describe images in more detail, we propose a framework for fine-grained pseudo tags, without the additional requirement of text-annotations and task extensions for the overlooked details. Inspired by the Xmodal-CTX (Kuo & Kira, 2022), which introduced a cross-modal retrieval module utilizing CLIP (Radford et al., 2021) to retrieve relevant text descriptions, providing complementary information to objects, we similarly construct relevant tags. Leveraging the data and training advantages of CLIP, we utilize it to extract visual features conditioned to generate the captions of the given image. A set of objects features, and grids features at different details levels extracted by CLIP-I are used. These features complement the visual information for both the main subject and background details with the advantages of preceding models. We construct rich detailed pseudo tags and employ asymmetric attention to adequately capture attention from the visual to the textual side. So that we can attain cross-modal understanding and fusion, which would support the task of generating captions with details.

However, directly utilizing the augmented image information is not sufficient to ensure the richness of the generated image captions. BLIP-2 (Li et al., 2023) and FUSECAP (Rotstein et al., 2024) employ frozen vision encoders in conjunction with powerful large language models to produce highly contextualized and descriptive captions. Nevertheless, the lack of explicit model guidance during the caption generation process makes it challenging to balance precision and richness within a unified end-to-end framework. To this end, we introduce a novel metric, Tags Coverage, to measure the level of details in cations. The constructed pseudo-tags and the metric are applied in NSC (Luo, 2020) for optimizing the reinforcement learning reward function (Rennie et al., 2017). So that, we enhance the richness and details in our generated captions without compromising much on their ac-

curacy. The main contributions of this paper are as follows, with a focus on balancing precision and richness in image captioning:

(1) *Enhanced Feature Representation for Image Captioning*: We leverage frozen pre-trained CLIP to construct pseudo tags, enabling the model to capture intricate image details. To support service-driven applications, we propose a multi-modal projector module that fuses feature information across modalities using asymmetric attention, enhancing descriptive precision and richness.

(2) *Richness-Oriented Evaluation Metric*: A new Tags Coverage (TC) metric is introduced to quantify caption granularity. TC is integrated into reinforcement learning, aligning training objectives by balancing accuracy and detail richness in generated captions.

(3) *Performance Gains*: Compared to baselines, our model achieves improvement in CIDEr, with a notable enhancement in TC over human annotations. These gains demonstrate our model's capability to deliver precise and richly detailed image captioning services.

## 2 RELATED WORK

### 2.1 ACCURACY AND FINE-GRAINED IN IMAGE CAPTIONING

Traditional metrics like BLEU (Papineni et al., 2002), ROUGE (Lin, 2004), METEOR (Banerjee & Lavie, 2005), CIDEr (Vedantam et al., 2015), and SPICE (Anderson et al., 2016) effectively assess the accuracy of image captioning by comparing generated captions with reference texts. Early image captioning models trained via time-wise cross-entropy performed well on these metrics but faced challenges like exposure bias (Ranzato et al., 2015) and misalignment between training and evaluation. Reinforcement learning approaches, including self-critical sequence training (Rennie et al., 2017), have mitigated these issues by using non-differentiable rewards such as CIDEr and BLEU, stabilizing reward variance and improving accuracy (Luo, 2020; Bujimalla et al., 2020). Despite these advances, models often fail to provide rich and flexible captions, overlooking important details in images. To address this, approaches like using image-text retrieval scores as rewards (Luo et al., 2018) and integrating CLIP-S (Hessel et al., 2021) have been explored to encourage distinct and descriptive captions. However, the lack of reference captions in these methods limits their granularity. In this study, we propose a service-oriented solution that introduces a novel metric, Tags Coverage (TC), to evaluate caption granularity. By leveraging CLIP (Radford et al., 2021), we retrieve fine-grained tags from images and use them as benchmarks to assess and enhance the richness of captions. This metric is incorporated into the reinforcement learning framework, aligning with service computing goals to balance descriptive precision and richness. Unlike previous methods, our approach eliminates reliance on extensive manual annotations or task extensions, ensuring scalability and adaptability for service applications like news reporting or assistive technologies. This balance between accuracy and granularity advances image captioning as a robust service, offering enhanced descriptive capabilities.

### 2.2 MODAL INTERACTION IN IMAGE CAPTIONING

Image captioning services rely on effective modal interaction to generate accurate and rich descriptions, a critical aspect of service computing. Earlier models, such as attention-based encoder-decoders (Gu et al., 2018; Wang et al., 2017; Lu et al., 2017), focused on feature extraction within visual and language domains. The introduction of object detectors (Anderson et al., 2018) significantly enhanced the identification of salient regions, improving caption accuracy. Advances in attention mechanisms (Vaswani et al., 2017; Huang et al., 2019) further enabled the use of Transformers for capturing global image contexts. $\mathcal{M}^2$Transformer (Cornia et al., 2020) utilized meshed memory storage to enhance encoder-decoder interactions, while the Multi-modal Transformer (Yu et al., 2019) unified intra- and inter-modal attention blocks. BLIP-2 (Li et al., 2023) and FUSECAP (Rotstein et al., 2024) employ frozen vision encoders in conjunction with powerful large language models to produce highly contextualized and descriptive captions. Building on recent advancements, our approach leverages frozen pre-trained large-scale models to extract detailed pseudo-labels, including object, grid, and tag features. The integration of asymmetric attention enables more effective mapping and fusion of these features, enriching the core image content with finer semantic details. Moreover, we use explicit model guidance during the caption generation process, which effectively balances precision and richness within a unified framework.

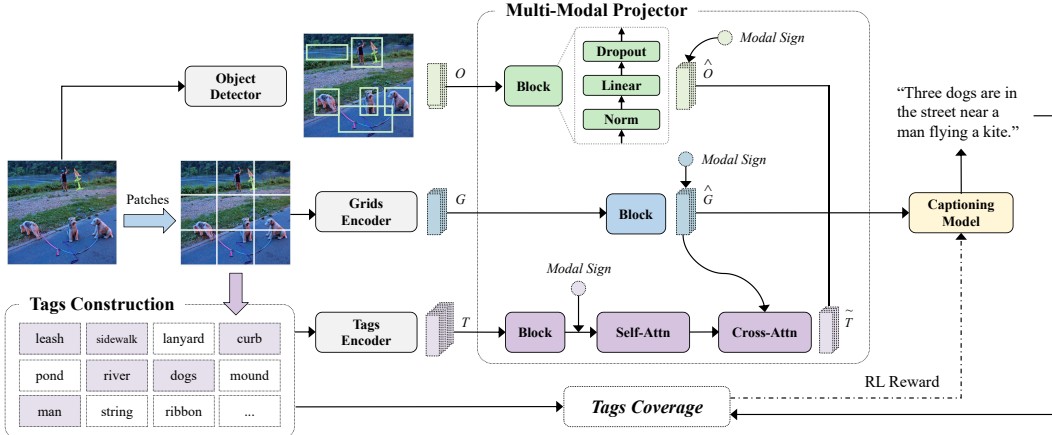

Figure 2: Overview of our proposed model architecture.

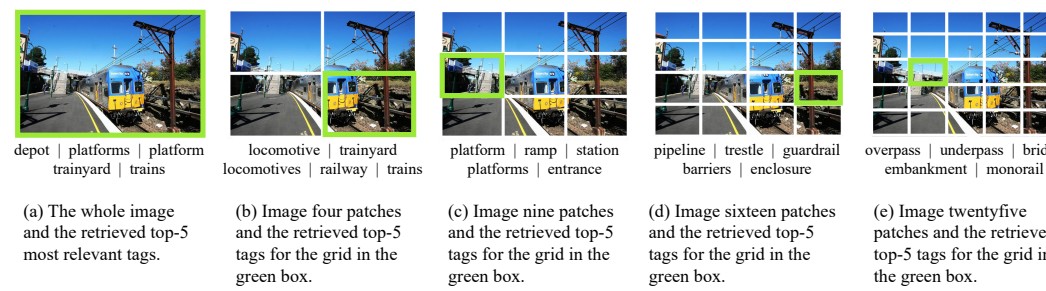

depot | platforms | platform trainyard | trains

locomotive | trainyard locomotives | railway | trains

platform | ramp | station platforms | entrance

pipeline | trestle | guardrail barriers | enclosure

overpass | underpass | bridge embankment | monorail

(a) The whole image and the retrieved top-5 most relevant tags.

(b) Image four patches and the retrieved top-5 tags for the grid in the green box.

(c) Image nine patches and the retrieved top-5 tags for the grid in the green box.

(d) Image sixteen patches and the retrieved top-5 tags for the grid in the green box.

(e) Image twentyfive patches and the retrieved top-5 tags for the grid in the green box.

Figure 3: Construct top-5 most relevant tags from different grids for (**a**) the whole image, (**b**) image four patches, (**c**) image nine patches, (**d**) image sixteen patches, (**e**) image twenty-five patches.

## 3 METHOD

We begin with an overview of the model architecture, followed by a introduction of the pseudo-tag construction module and the multi-modal projector. Next, we present the optimized reinforcement learning strategy based on NSC (Luo, 2020), where the proposed Tags Coverage metric is employed as the reward to enhance caption quality.

### 3.1 MODEL ARCHITECTURE

As shown in Fig. 2, it consists of multi-modal extractors, a multi-modal projector, and a captioning model. We leverage the advantages of existing cross-modal large-scale models to perform features extractors for objects, grids, and tags within images. Specifically, following the methodology proposed in Xmodal-CTX (Kuo & Kira, 2022), we use the same objects features extracted by the object detector pre-train on Visual Genome (Krishna et al., 2017). Employing frozen CLIP-I and CLIP-T from different branches of CLIP (Radford et al., 2021), we extract features for image grids and construct fine-grained pseudo tags. Subsequently, we design a multi-modal projector aiming to fuse the extracted features, thoroughly utilizing self-attention mechanisms and cross-attention mechanisms for sampling tags features. Finally, a captioning model $\mathcal{M}^2$Transformer (Cornia et al., 2020) is connected and the fusion of cross-modal features is fed in. The tags constructed before are used for calculating the Tags Coverage metric, which, in turn, will be employed for model fine-tuning, encouraging the generation of image captions containing a more comprehensive set of tags from the tags repository.

### 3.2 TAGS CONSTRUCTION

The pseudo tags are proposed to enable the model to learn from, beyond human annotations but reflecting images realistically. And it encourages the model to learn without the necessity of addi-

tional text-annotations and other vision-language tasks during training. We similarly transform the tags construction problem into a cross-modal retrieval task, performing specific retrieval for each sub-region of images from the tags repository. Our goal is to focus on the details of images from different levels. Through this approach, we can focus on different local regions of the image, thereby perceiving more detailed local information.

The first step is to construct a tags repository for each image. While existing different datasets contain different annotations for images, to ensure the consistency of the dataset and leverage the enriched knowledge obtained by large-scale models, we create the tags repository from the ground truth annotations within the COCO dataset (Lin et al., 2014) (It is also used in our experiments). After tokenization, deduplication, and cleaning processes, the tags repository is formed using various vocabulary tokens. Employing the text branch CLIP-T, we encode the tags from the constructed tags repository as tags repository features $T*$.

Then, we can retrieve the corresponding tags from different features of sub-regions within images, based on the cross-modal joint embedding of CLIP. CLIP trained on vast datasets and aligns images and texts through contrastive learning to ensure textual and visual consistency. We explicitly consider dividing the image $I$ into one, four, nine, sixteen, and twenty-five patches as varying region sizes(finer partitions), which would impact the level of details within the image. Using CLIP-I image branch, we encode the patches of the image $I$ into grid features $G = \{G^{\#1}, G^{\#4}, G^{\#9}, G^{\#16}, G^{\#25}\}$, where $G^{\#i} = \{g_j^{\#i}|j \in \{1, 2, ..., i\}\}$, and $g_j^{\#i}$ denotes the $j$-th grid feature from $i$ sub-patch(es) in the Image $I$ where each $g \in \mathbb{R}^{d_g}$. Utilizing the grids features $G^{\#i}$ as queries and tags repository $T*$ as retrieval keywords, we retrieve the top-$k$ tags according to the highest cosine similarity scores, use CLIP-T text branch and obtain the tags features $T = \{T^{\#1}, T^{\#4}, T^{\#9}, T^{\#16}, T^{\#25}\}$, where $T^{\#i} = \{t_{j,k}^{\#i}|j \in \{1, 2, ..., i\}, k \in \{1, 2, ..., \text{top-}k\}\}$, and $t_{j,k}^{\#i}$ denotes the $k$-th tag feature in top-$k$ tags features corresponding to the $j$-th grid in $i$ patch(s) where each $t \in \mathbb{R}^{d_t}$. Some examples of the top-5 results are shown in Fig. 3.

After processing mentioned above, we obtain the corresponding tags for each image patch. These tags will be utilized in two aspects: one will be fed into the model for joint training within the multi-modal projector, and the other will be used in our proposed metric TC for calculating.

## 3.3 MULTI-MODAL PROJECTOR

The multi-modal projector aims to map inputs from different modalities into a shared representation space to achieve information fusion for cross-modalities. In our work, we utilize frozen pre-trained models CLIP (Radford et al., 2021) to extract features from image $I$, including grids features $G$ and tags features $T$. Let $O = \{o_1, o_2, ..., o_n\}$, where each $o \in \mathbb{R}^{d_o}$, represents a set of number $n$ objects features $o_i$ detected by a frozen pre-trained object detector. Within the multi-modal projector, we use a Block including Norms, Fcs, and Drops to map features from different modalities to adapt to downstream sequence generation, which can be modeled as:

$$\hat{O} = [\text{Drop}(\text{FC}(\text{Norm}(O))), m_O],$$
$$\hat{G} = [\text{Drop}(\text{FC}(\text{Norm}(G))), m_G], \tag{1}$$
$$\hat{T} = [\text{Drop}(\text{FC}(\text{Norm}(T))), m_T],$$

where Norm denotes the normalization layer, FC denotes the fully connected layer, Drop refers to the dropout layer, $[\cdot, \cdot]$ is concatenation and $m$ denotes a sign from different modals of objects, grids, and tags. In our works, distinct FC layers are employed to encode various features, addressing different modalities and granular levels.

To thoroughly explore inter-modal information and enhance cross-modal integration for the correlation between grids and tags, we introduce asymmetric attention with summed shortcut connections and design a self-attention layer (SA) and a cross-attention layer (CA). Specifically, the tags features $\hat{T}$, processed through a Block, are input into the self-attention layer to obtain $\hat{T}_{\text{SA}}$. Subsequently, grid features $\hat{G}$ are injected into the tag features $\hat{T}_{\text{SA}}$ via the cross-attention layer to produce $\hat{T}_{\text{CA}}$. This result is then fed into the Feed-Forward Network (FFN), where it is added to the output of the cross-attention $\hat{T}_{\text{CA}}$, generating the visual-perceptive tag representation $\widetilde{T}$:

$$\hat{T}_{\text{SA}} = \text{Norm}(\text{SA}(\hat{T}) + \hat{T}), \tag{2}$$

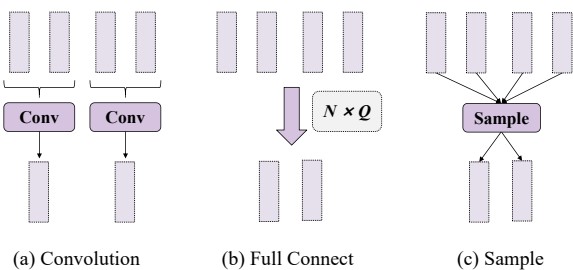

(a) Convolution          (b) Full Connect          (c) Sample

Figure 4: Three types of refinements in cross attention.

$$\hat{T}_{\text{CA}} = \text{Norm}(\text{CA}(\hat{T}_{\text{SA}}, \hat{G}) + \hat{T}_{\text{SA}}), \tag{3}$$

$$\widetilde{T} = \text{Norm}(\text{FNN}(\hat{T}_{\text{CA}}) + \hat{T}_{\text{CA}}), \tag{4}$$

where FFN consists of two linear layers, a ReLU layer, and a Dropout layer.

It is noteworthy that when computing cross-attention, we do not use the entire $\hat{T}_{\text{SA}}$ as the query $Q$. We explicitly guide the model to learn fine-grained tags from different levels, but it does not imply that every tag is beneficial for the model. Meanwhile, to reduce the computational cost, we perform refine the tags features after self-attention $\hat{T}_{\text{SA}*}$ by three ways in Fig. 4. We envision utilizing a fully connected layer (FC) for a linear transformation of all tag features, employing sampling (SP) to extract features from certain tags features, and leveraging convolution (CN) to capture local information of tags features. We employ the scaled dot-product attention, which operates on three sets of vectors. Based on the similarity distribution between the query and key, we calculate the weighted sum of the value:

$$
\begin{aligned}
Q_{\text{FC}} &= W_q \text{FC}(\hat{T}_{SA}), \\
Q_{\text{SP}} &= W_q \text{SP}(\hat{T}_{SA}), \\
Q_{\text{CN}} &= W_q \text{CN}(\hat{T}_{SA}),
\end{aligned}
\tag{5}
$$

$$\text{CA}(Q, K, V) = \text{softmax}\left(\frac{QK^T}{\sqrt{d}}\right)V, \tag{6}$$

where $W_q$ is learnable weights, $Q \in \{Q_{\text{FC}}, Q_{\text{SP}}, Q_{\text{CN}}\}$ is a matrix of tags features after refinements $\hat{T}_{\text{SA}*}$, $K$ and $V$ both contain $\hat{G}$ keys and values, and $d$ is a scaling factor. Subsequently, the mapped objects features $\hat{O}$, grids features $\hat{G}$, and tags features $\widetilde{T}$ are concatenated along the sequence dimension to form $z = [\hat{O}, \hat{G}, \widetilde{T}]$, which is fed into a captioning model for sequence-to-sequence learning.

## 3.4 TAGS COVERAGE METRIC

The prevailing metric for assessing the accuracy of generated image captions is CIDEr (Vedantam et al., 2015), which calculates the cosine similarity between the generated and reference sentences based on N-gram. However, higher CIDEr scores may result in generated sentences that are relatively short and lack details. To assess the granularity of generated sentences, one straightforward approach is to examine whether the output captions contain more detailed information. Inspired by (Zhao et al., 2019), we propose a novel metric Tags Coverage (TC), employing the fine-grained tags construction in Section 3.2 as the details benchmark. This metric serves as a precision-like score in Eq. equation 7, quantifying the proportion of tags tokens within the tokens of the output sentence. For each image, we utilize the top-100 tags with the highest cosine similarity which is outlined in the Tags Construction section, as a reference for assessment.

$$\text{TC} = \frac{|\{\text{tags tokens}\} \cap \{\text{caption tokens}\}|}{|\{\text{caption tokens}\}|}. \tag{7}$$

This metric is not independently assessed but relies on the prior knowledge of the pre-trained CLIP (Radford et al., 2021). The accuracy of the pseudo tags constructed by the pre-trained model CLIP

determines the realism of the granularity reflected by TC. However, it is essential to note that covering more tags in the generated sentences does not necessarily mean better quality; the inclusion of function words is also necessary for organizing sentence structures to ensure coherence and fluency. Additionally, some tags that may be misidentified by CLIP are also incorporated into the tags set. It effectively increases the fault tolerance of TC as humans also make mistakes sometimes. Therefore, TC may not reach an exceptionally high score (At least it won't get 100%). As we generate sentences that encompass more tags, the TC score increases, while the CIDEr score may decrease. It serves the purpose of ensuring that generated captions contain more details and should be considered alongside the CIDEr metric to evaluate sentence quality comprehensively. We apply NSC (Luo, 2020), a various form of self-critical sequence training (Rennie et al., 2017). During the reinforcement learning fine-tuning stage, we consider incorporating TC as a reward to balance the accuracy and granularity of the output captions:

$$
\begin{aligned}
R_{\text{CIDEr}} &= \text{CIDEr}(\hat{y}), \\
R_{\text{TC}} &= \text{TC}(\hat{y}),
\end{aligned}
\tag{8}
$$

$$
R(\hat{y}) = \lambda_1 R_{\text{CIDEr}} + \lambda_2 R_{\text{TC}},
\tag{9}
$$

$$
\nabla_\theta L(\theta) \approx -(R(\hat{y}) - b)\nabla_\theta \log p(\hat{y}|z),
\tag{10}
$$

where $R(\cdot)$ is the rewards function, $\hat{y}$ is a sampled caption, $b$ denotes the reward of the baseline for $\hat{y}$ acquired by sampling, and $z$ is the latent variable in Section 3.3. The CIDEr reward is calculated according to the ground-truth sentence $y^s$, which encourages the model to preserve the content in the input image. The TC reward employs the above-mentioned in Eq. equation 7, encouraging the model to generate finer-grained information from the image. The coefficients $\lambda_1$ and $\lambda_2$ denote the weights of the two components, which are tunable hyper-parameters. The combined effect of both ensures the accuracy and details.

In addition, as discussed above, the quality of the tags generated by CLIP affects the model's caption generation performance. To address this, we further refine Equation (9) by incorporating the correlation between the generated tag tokens and the caption tokens when computing the tag consistency (TC). The revised formulation is as follows:

$$
R'_{\text{TC}} = \text{TC}(\hat{y}) \times S(\text{B(caption tokens)}, \text{B(tags tokens)}),
\tag{11}
$$

where $S(\cdot)$ denotes the cosine similarity, and $B(\cdot)$ refers to the BERT-Small version (Turc et al., 2019) used for obtaining vector representations. $R'_{\text{TC}}$ aims to allow the model to simultaneously consider the semantic consistency between the CLIP-generated caption and the ground-truth caption, as well as the richness of the generated caption.

### 3.5 COMPLEXITY ANALYSIS

In comparison to models with only a single encoder, although our approach introduces two encoders (grid encoder and tags encoder), the overall number of training parameters and duration has not increased significantly. This is attributed to the fact that the image segmentation and tag generation we employ are based on pre-trained large models. The operations of the two encoders can be completed during the preprocessing stage of our model, eliminating the need for redundant computations during training. In terms of the complexity of training parameters, the added parameter quantity is 768 x 512 for the grid part and 512 x 512 for the tag part (specific experimental parameters are detailed in Section 4.3). Hence, the complexity of our proposed method remains manageable.

## 4 EXPERIMENTS

### 4.1 DATASETS AND METRICS

We train and evaluate our model on the widely used benchmark MS-COCO (Lin et al., 2014) in the field of image captioning. For a fair comparison, we adopted the split method proposed by Karpathy (Karpathy & Fei-Fei, 2015), where each image is associated with a minimum of five manual annotations captions. Of these, 5000 images were used for validation, another 5000 for testing, and the rest for training. According to the standard evaluation have been proposed, mostly based on comparing generated captions to human ones, we utilized a comprehensive set of captioning metrics: BLEU (Papineni et al., 2002), METEOR (Banerjee & Lavie, 2005), ROUGE (Lin, 2004), CIDEr (Vedantam et al., 2015), SPICE (Anderson et al., 2016), along with the proposed TC.

| Models | B-1 | B-4 | M | R | C | S | TC |
|---|---|---|---|---|---|---|---|
| Att2in (Rennie et al., 2017) | 78.3 | 35.6 | 27.3 | 56.9 | 119.2 | 20.6 | 18.7 |
| Up-Down (Anderson et al., 2018) | 79.6 | 36.8 | 28.0 | 57.7 | 121.9 | 21.4 | 19.1 |
| Transformer (Vaswani et al., 2017) | 80.7 | 38.9 | 29.0 | 58.7 | 129.2 | 22.8 | 19.6 |
| $\mathcal{M}^2$ (Cornia et al., 2020) | 80.8 | 39.1 | 29.1 | 58.4 | 131.2 | 22.6 | 19.5 |
| Xmodal-CTX (Kuo & Kira, 2022) | 81.2 | 40.0 | 29.2 | 59.3 | 133.2 | 22.8 | 20.0 |
| HAAV (Kuo & Kira, 2023) | 80.3 | 37.1 | 28.2 | 58.0 | 127.4 | 21.8 | 19.3 |
| BLIP-2 (Li et al., 2023) | 81.4 | 39.5 | 29.6 | 59.8 | 133.6 | 22.3 | 21.7 |
| FUSECAP (Rotstein et al., 2024) | 80.5 | 39.2 | 29.2 | 59.5 | 133.1 | 21.9 | 22.5 |
| Human annotations | / | / | / | / | / | / | 17.7 |
| Ours$_{(CIDEr)}$ | 81.5 | 39.9 | 29.5 | 59.4 | 134.1 | 23.3 | 20.0 |
| Ours$_{(CIDEr+R_{TC})}$ | 81.1 | 39.4 | 29.1 | 59.1 | 132.2 | 23.2 | 23.3 |
| Ours$_{(CIDEr+R'_{TC})}$ | **81.9** | **40.6** | **30.2** | **60.5** | **135.2** | **23.7** | **23.9** |

Table 1: Comparison with baselines for image captioning results on the test set of MS-COCO Karpathy split. Human annotations refer to the ground-truths. B-$N$, M, R, C, S, and TC represent BLEU@$N$, METEOR, ROUGE-L, CIDEr, SPICE, and Tags Coverage metrics.

| Strategy | B-1 | B-4 | M | R | C | S | TC |
|---|---|---|---|---|---|---|---|
| XE | 76.7 | 35.0 | 28.3 | 56.8 | 117.8 | 21.7 | 20.3 |
| NSC$_{(w/ CIDEr)}$ | 81.5 | 39.9 | 29.5 | 59.4 | 134.1 | 23.3 | 20.0 |
| NSC$_{(w/ CIDEr+R_{TC})}$ | 81.1 | 39.4 | 29.1 | 59.1 | 132.2 | 23.2 | 23.3 |
| NSC$_{(w/ CIDEr+R'_{TC})}$ | **81.9** | **40.6** | **30.2** | **60.5** | **135.2** | **23.7** | **23.9** |

Table 2: The results of our models with and without the reward of TC on the test set. XE and NSC mean the cross-entropy loss training and new self-critical finetuning.

## 4.2 SELECTED BASELINES

We select the following baselines for comparison: Att2in (Rennie et al., 2017): It introduces a reinforcement learning method for optimizing image captioning systems. It directly optimizes the CIDEr metric, making it highly effective for image captioning tasks. Up-Down (Anderson et al., 2018): It proposes a novel combined bottom-up and top-down attention mechanism, achieving state-of-the-art performance in image captioning and VQA tasks. Transformer (Vaswani et al., 2017): It is a based model for image caption tasks. $\mathcal{M}^2$ (Cornia et al., 2020): It introduces a meshed transformer with memory for improved image captioning, achieving state-of-the-art performance. Xmodal-CTX (Kuo & Kira, 2022): It introduces a novel approach in visual captioning, utilizing auxiliary inputs to capture missing information and improving model grounding. HAAV (Kuo & Kira, 2023): It proposes an innovative image captioning approach that treats various visual and textual encodings as augmented views of the input image. BLIP-2 (Li et al., 2023): It bridges frozen image encoders and large language models using a lightweight Querying Transformer. FUSECAP (Rotstein et al., 2024) is a data-centric approach that enriches generic image captions by fusing outputs from frozen vision experts with original captions using a large language model.

## 4.3 EXPERIMENT SETTINGS

We tune the top-k parameter on the validation set and find that the performance saturates at $k = 9$ in the tag construction. The objects features we use follow Xmodal-CTX (Kuo & Kira, 2022). The grids and tags features extractors we used are frozen pre-trained CLIP. The dimensions of the extracted object features, grid features, and tag features are 2054, 768, and 512 respectively. After the multi-modal projector, they are fused to the latent variable $z$ with a dimension of 512. We train our model with Adam optimization and the Reduce-LR-On-Plateau method on A800. The model is initially trained using cross-entropy XE for 25 epochs with the learning rate of $1 \times 10^{-4}$, followed by fine-tuning with NSC (Luo, 2020) reinforcement learning with appropriate rewards for another 15 epochs. All comparisons among experimental methods were conducted under fair conditions. More experimental details are provided in the Appendix section.

## 4.4 COMPARISON FOR IMAGE CAPTIONING RESULTS

Firstly, we compare our model with the trained-from-scratch methods as shown Table 1. Except for $\mathcal{M}^2$ (Cornia et al., 2020), which is obtained from their published model, the other models are

| Asymmetric Attention | XE | NSC | B-1 | B-4 | C | S | TC |
|---|---|---|---|---|---|---|---|
| w/o attn | ✓ | | 76.9 | 34.7 | 115.8 | 21.3 | 20.3 |
| w/ attn | ✓ | | 76.7 | 35.0 | 117.8 | 21.7 | 20.3 |
| w/ multi-attn | ✓ | | 77.3 | 35.3 | 118.4 | 21.9 | 20.2 |
| w/ attn | | ✓ | 81.9 | 40.6 | 135.2 | 23.7 | 23.9 |
| w/ multi-attn | | ✓ | **82.3** | **41.2** | **135.9** | **24.1** | **24.3** |

Table 3: Ablation study for the proposed asymmetric attention in the multi-modal projector. The first row without attention means features after extractors are directly fully connected to the next part in the model. The multi-attn refers to multi-head attention with heads of 8 here.

reproduced in the codes framework (Luo et al., 2018). We mark the best scores in bold and the second with the underline. Our method achieves the best overall performance across all metrics, including 135.2 CIDEr, 23.7 SPICE, and 23.9 TC. Compared to the strongest baseline FUSECAP, our model improves CIDEr by 2.1, SPICE by 1.8, and TC by 1.4. It also outperforms BLIP-2 by 1.6 in CIDEr, 1.4 in SPICE, and 2.2 in TC, while maintaining stronger or comparable performance on BLEU, METEOR, and ROUGE-L metrics.

Secondly, we observe that $Ours_{(CIDEr)}$ performs well on standard evaluation metrics, achieving 134.1 in CIDEr and 23.3 in SPICE, while maintaining a moderate TC score of 20.0. This outcome aligns with its design objective of prioritizing caption accuracy, though it may generate less detailed descriptions. In comparison, $Ours_{(CIDEr+R_{TC})}$ shows a slight decrease in CIDEr (132.2) and SPICE (23.2), but achieves a higher TC score, 3.3 points above FUSECAP and 5.6 points above human annotations, suggesting improved caption richness. Furthermore, $Ours_{(CIDEr+R'_{TC})}$ achieves the highest scores across all metrics, indicating that jointly optimizing for semantic consistency and richness leads to more balanced captions in terms of both accuracy and detail. These results demonstrate the effectiveness of incorporating a dual-objective reward in guiding caption generation.

## 4.5 STRATEGY ANALYSIS OF THE PROPOSED METHOD

From the results in Table 2, it is observed that the model optimized through CIDEr reward may incur some losses in terms of fine-grained metric. Specifically, to achieve higher CIDEr scores, the model tends to generate results closer to the ground truths, limiting the generated vocabulary within a certain range defined by human annotations. Our TC reward encourages the model to explore more possibilities, leading to a broader coverage of tags details in the generated vocabulary. Thus, using TC reward does help to cover more detailed tags in image captioning.

## 4.6 ABLATION STUDY

We conducted an ablation analysis on our asymmetric attention multi-modal projector, as shown in the first row "w/o attn" of Table 3. In this setting, we directly input the extracted objects features, grids features, and tags features through the Block into $\mathcal{M}^2$ for training. It can be observed that the multi-modal projector with attention improves 2.0 on CIDEr when using the strategy of XE. After applying the asymmetric attention projection across modalities, features from different modalities can better integrate information to meet the requirement of the downstream task of generating more fine-grained descriptions. To verify it, we further enhanced attention by incorporating a multi-head attention mechanism with $h = 8$. Compared with attention, we observe a further performance improvement - 0.7 in CIDEr and 0.4 in TC with multi-head attention after reinforcement learning fine-tuning, indicating the effectiveness of our multi-modal projector. Due to the higher computational cost of multi-head attention mechanism, we did not opt for this set.

## 5 CONCLUSION

In this paper, we address the challenge of balancing precision and richness in image captioning. We propose a CLIP-based model that captures fine-grained tags by extracting object, grid, and tag features and integrating them with an asymmetric attention projector. To encourage accurate yet detailed captions, we introduce TC, a fine-grained evaluation metric, into the reinforcement learning reward. Experiments show that our method generates captions with both precision and richness.

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

| Dimension Reduction | B-1 | B-4 | C | S | TC |
|---|---|---|---|---|---|
| CN | 76.0 | 34.0 | 114.0 | 21.1 | **20.4** |
| FC | 76.7 | **35.0** | **117.8** | **21.7** | 20.3 |
| SP | **77.1** | 35.1 | 116.9 | 21.4 | 20.3 |

Table 4: Different ways of dimension reduction in asymmetric cross attention. CN, FC, and SP refer to convolution, fully connect, and sample, see text for detailed descriptions of three ways.

# A  APPENDIX

Several additional experiments and analyses are provided in the appendix, including different refinements on tag features, studies on the balance between accuracy and fine-grained rewards, evaluations under varying top-$k$ settings, and qualitative results with visualizations comparing model outputs and human annotations. We commit to releasing the complete version of the dataset and code at the acceptance stage.

## A.1  DIFFERENT TYPES OF REFINEMENTS ON TAGS FEATURES

We performed refinements on the processed tags features to reduce computational costs and enhance attention efficiency. Specifically, we explored three dimension reduction strategies in the asymmetric cross-attention module: convolution (CN), fully connected (FC), and sampling (SP). Table 4 presents the performance comparison across these approaches using standard evaluation metrics.

The FC method shows the highest performance in CIDEr (117.8), SPICE (21.7), and BLEU-4 (35.0), indicating its effectiveness in integrating information from high-dimensional tag features. This can be attributed to the capacity of fully connected layers to model comprehensive feature interactions, which contributes to improved caption accuracy. The SP method achieves slightly lower scores than FC in CIDEr and BLEU-4 but attains the highest BLEU-1 score (77.1), suggesting it may help preserve more diverse n-gram expressions by selectively retaining tag features. Although its TC score (20.3) is marginally lower than that of CN (20.4), the performance remains competitive. The CN approach yields the lowest CIDEr score (114.0) among the three, but it performs best in the TC metric (20.4), implying that convolutional operations may help retain fine-grained details relevant to tag coverage, despite producing slightly less accurate captions overall.

In general, FC is recommended when accuracy is prioritized in caption generation, while SP can be a viable option for scenarios that emphasize diversity and flexibility. CN serves as a balanced alternative, offering modest performance in both accuracy and richness.

## A.2  BALANCE WEIGHTS BETWEEN ACCURACY AND FINE-GRAINED DESCRIPTIONS

We examine the impact of varying the reward weights between CIDEr and TC ($R_{TC}$ and $R'_{TC}$) during model fine-tuning, as shown in Table 5.

When the weight of TC increases while keeping the CIDEr weight fixed at 1, a moderate trade-off is observed: the CIDEr score decreases slightly (e.g., from 134.1 to 132.2 with $R_{TC}$), while the TC score improves notably (from 20.0 to 23.3). Besides, in the case of the refined reward $R'_{TC}$, both accuracy and richness improve together, with CIDEr increasing to 135.2 and TC reaching 23.9 when the weights are balanced. These findings indicate that incorporating both semantic consistency and richness into the reward formulation can help produce captions that maintain accuracy while providing more detailed descriptions. Careful tuning of reward weights is therefore important for achieving a balance between precision and informativeness in image captioning.

## A.3  TOP-K EXPERIMENTAL RESULTS

We further evaluated our model by varying the hyper-parameter top-$k$ in *cross-entropy training (XE)*. As depicted in Table 7, the optimal results in terms of CIDEr, SPICE, and Tags Coverage were achieved when utilizing top-9 tags features as the parameter. The accuracy of generated captions, measured by metrics like CIDEr, gradually improved with an increase in top-$k$ and reached optimal

| Weight of CIDEr | Weight of $R_{TC}$ | B-1 | B-4 | C | S | TC |
|---|---|---|---|---|---|---|
| 1 | 0 | 81.5 | 39.9 | **134.1** | **23.3** | 20.0 |
| 1 | 0.5 | **81.6** | **40.0** | 133.3 | 23.2 | 21.3 |
| 1 | 1 | 81.1 | 39.4 | 132.2 | 23.2 | **23.3** |
| Weight of CIDEr | Weight of $R'_{TC}$ | B-1 | B-4 | C | S | TC |
| 1 | 0 | 81.5 | 39.9 | 134.1 | 23.3 | 20.0 |
| 1 | 0.5 | 81.7 | 40.1 | 134.7 | 23.4 | 22.5 |
| 1 | 1 | **81.9** | **40.6** | **135.2** | **23.7** | 23.9 |

Table 5: Image captioning results of different rewards on MS-COCO Karpathy test split.

| Hyperparameter | XE | NSC | Note |
|---|---|---|---|
| num_layers | 3 | * | Number of encoder and decoder layers |
| encoder_size | 512 | * | encoder embedding size |
| ff_size | 2048 | * | feed-forward network size |
| h | 8 | * | head of multi-head attention |
| dropout_rate | 0.1 | * | dropout rate |
| max_lenth | 20 | * | max length of captions |
| vocab_size | 9487 | * | length of vocabulary |
| max_epochs | 25 | +15 | trainning epochs |
| lr | 1e-4 | 5e-6 | learning rate |
| optimizer | Adam | * | Adam optimizer |
| objs_size | 2054 | * | objects features size |
| grids_size | 768 | * | grids features size |
| tags_size | 512 | * | tags features size |
| tags_num | 100 | * | number of tags to calculate Tags Coverage |

Table 6: Hyperparameters for cross entropy (XE) training and new self-critical (NSC) training. The values for untuned parameters are inherited from the base image captioning model.

| | B-1 | B-4 | M | R | C | S | TC |
|---|---|---|---|---|---|---|---|
| 3 | 76.9 | 34.6 | 27.9 | 56.6 | 116.1 | 21.2 | 20.3 |
| 4 | 76.6 | 34.5 | 27.9 | 56.5 | 116.3 | 21.5 | 20.2 |
| 5 | 76.6 | 34.5 | 28.0 | 56.6 | 116.3 | 21.4 | 20.5 |
| 6 | 76.8 | 34.9 | 27.9 | 56.8 | 116.7 | 21.7 | 20.2 |
| 7 | 76.8 | 34.6 | 27.9 | 56.6 | 115.9 | 21.5 | 20.2 |
| 8 | 76.9 | 34.4 | 27.9 | 56.5 | 115.9 | 21.4 | 20.5 |
| 9 | 76.7 | 35.0 | 28.3 | 56.8 | 117.8 | 21.7 | 20.3 |
| 10 | 76.9 | 35.0 | 28.0 | 56.5 | 117.1 | 21.4 | 20.3 |
| 11 | 77.1 | 35.0 | 28.1 | 56.8 | 117.1 | 21.5 | 20.2 |
| 12 | 76.7 | 34.9 | 27.8 | 56.6 | 115.7 | 21.3 | 20.0 |

Table 7: Captioning results (trained with XE loss) of our model with top-$k$ tags features used: from top-3 to top-12. B-$N$, M, R, C, S, and TC represent BLEU@$N$, METEOR, ROUGE-L, CIDEr, SPICE, and Tags Coverage metrics.

performance around top-9. Therefore, in the main paper, we conducted experiments using top-9 as our parameter of the number of tags features used in our model. The table illustrates that, as the number of tag features increases, the model can glean more knowledge from the rich tag information, thereby enhancing the accuracy of the model in generating captions and improving the Tags Coverage metric. However, it's crucial to note that a higher top-$k$ parameter is not necessarily better. On one hand, we cannot ensure that all tags inputted into the model are perfectly accurate,

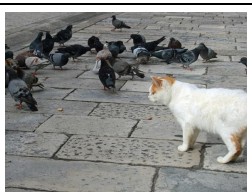

**Ours**
A cat standing on a sidewalk next to a flock of pigeons
**Human**
A very cute cat near a bunch of birds

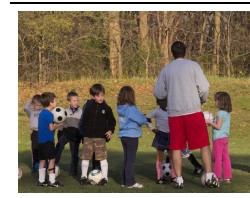

**Ours**
A group of children standing in a field with soccer balls
**Human**
A group of young children standing around a field

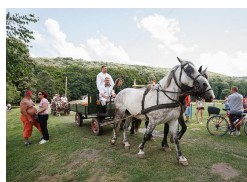

**Ours**
A white horse pulling a carriage with people in the grass
**Human**
A horse and buggy that is on a grassy field

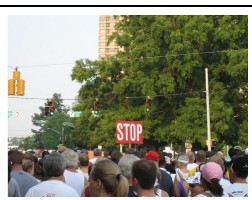

**Ours**
A crowd of people walking down a street with a stop sign
**Human**
People stand in a city street at a rally

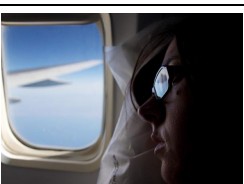

**Ours**
A woman wearing sunglasses looking out of an airplane window
**Huaman**
A woman sleeping on a plane with a window view of the wing

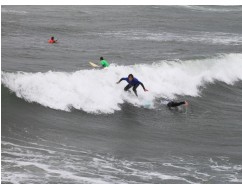

**Ours**
A group of surfers riding a wave on surfboards in the ocean
**Human**
A person is riding the waves on a surfboard.

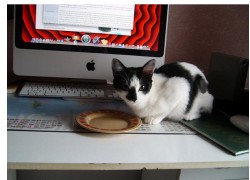

**Ours**
A black and white cat sitting on a desk next to a computer
**Human**
A can laying on a desk in front of a computer

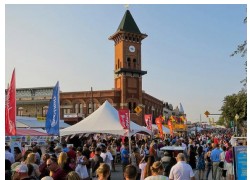

**Ours**
A crowd of people walking in front of a building with a clock tower
**Human**
A crowd of people walking in an outdoor fair

Figure 5: Examples of image captioning services generated by our model and the ground-truths.

as label construction may have inherent errors. On the other hand, an excess of tags may introduce noise, preventing the model from focusing on the essential information it should learn.

### A.4 IMPLEMENTATION DETAILS

We provide a list of hyper-parameters including their values during cross-entropy training(XE) and new self-critical training (NSC) in Table 6. Others not present are following the works before. For cross-entropy training, the model can be trained with three Nvidia 3090 GPUs. For NSC training, the model can be trained with a single A800 GPU.

### A.5 QUALITATIVE RESULTS AND VISUALIZATION

Fig. 5, Fig. 6 and Fig. 7 present the qualitative results obtained by our model and the original human annotations. The portions of the captions that subjectively represent detailed tags information are highlighted in purple and underlined. For instance, the colors of the horse and the cat in images are caught by our model. On average, our model covers more fine-grained details and object relationships, producing descriptions with both high accuracy and details.

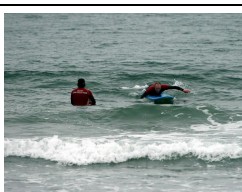

**Ours**
Two people in the ocean one is riding a wave on a surfboard
**Human**
A person riding a surf board on a body of water

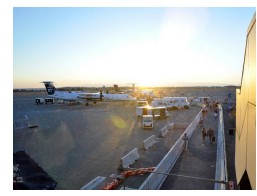

**Ours**
A large airplane parked at an airport with people walking by
**Human**
A airplane that is sitting on a tarmac

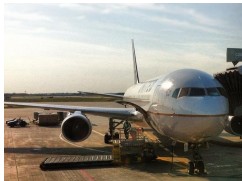

**Ours**
A large passenger jet sitting on top of an airport tarmac
**Human**
An airplane parked on the tarmac at an airport

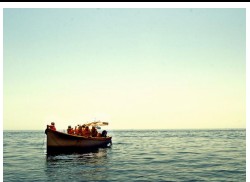

**Ours**
A boat filled with people floating on top of a body of water
**Human**
A bunch of people are on a small boat

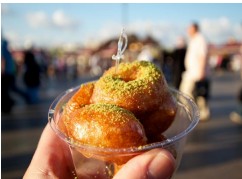

**Ours**
A person holding a glass filled with sugar covered donuts
**Human**
A cup of food in a persons hand

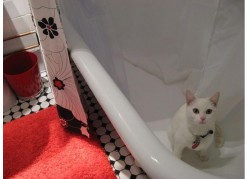

**Ours**
A white cat sitting in a bathtub next to a red and white rug
**Human**
A cat sitting in a bathtub behind the curtain

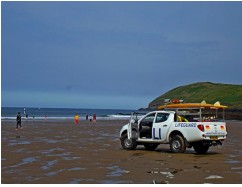

**Ours**
A truck parked on the beach with people walking on the sand
**Human**
A live guard truck parked on a beach

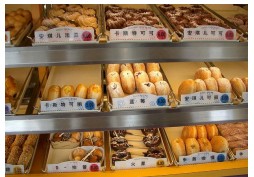

**Ours**
A display case in a bakery filled with donuts and pastries
**Human**
a bakery with boxes of donuts and bread

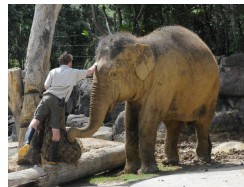

**Ours**
A man is petting an elephant that is standing next to a log
**Human**
A man is reaching over to an elephant

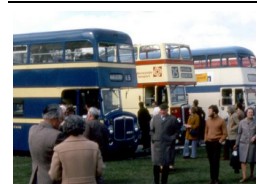

**Ours**
A group of people standing outside of a double decker bus
**Human**
A group of people are on the grass by busses

Figure 6: Examples of image captioning services generated by our model and the ground-truths.

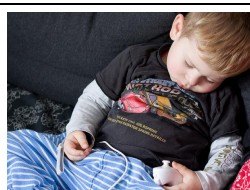

**Ours**
A young boy laying on a couch holding a nintendo wii game controller

**Human**
A sleeping child holding a Wii controller in hand

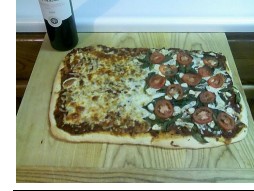

**Ours**
A pizza sitting on a wooden cutting board with a bottle of wine

**Human**
A pizza sitting on top of a wooden cutting board

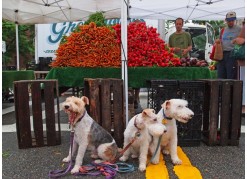

**Ours**
Three dogs sitting next to a fruit stand with fruits and vegetables

**Human**
Three dogs sitting side by side in the street

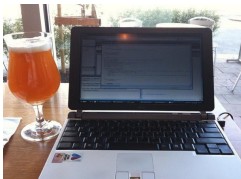

**Ours**
A laptop computer sitting on a desk with a glass of orange juice

**Human**
A laptop on a wooden table near a drink

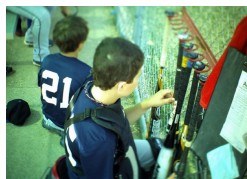

**Ours**
Two young boys sitting in a baseball dugout with baseball bats

**Huaman**
A couple of men standing next to each other

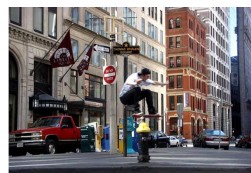

**Ours**
A man jumping in the air on a skateboard over a fire hydrant

**Human**
A man that is in the air with a skateboard

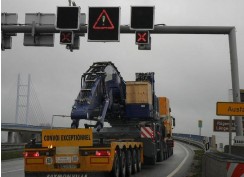

**Ours**
A construction truck with two traffic lights on a bridge

**Human**
A street scene with a large struck driving by

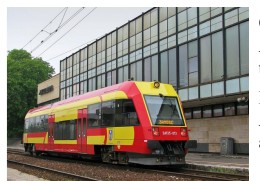

**Ours**
A red and yellow train on the tracks in front of a building

**Human**
A photo of a train passing by a building

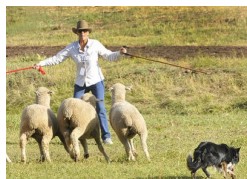

**Ours**
A woman standing in a field with a herd of sheep and a dog

**Human**
A man is with some sheep in a field

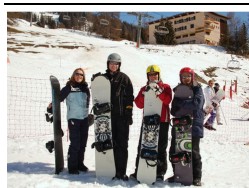

**Ours**
A group of people standing in the snow holding snowboards

**Human**
A group of four people standing next to each other in the snow

Figure 7: Examples of image captioning services generated by our model and the ground-truths.

