# OpenReview forum: "Balancing Precision and Richness in Image Caption Services for Enhanced Descriptive Accuracy"
_ICLR.cc/2026/Conference — ICLR 2026 Conference Withdrawn Submission_

### Official Review · Reviewer_R7ub · 2025-10-30

**Soundness:** 3
**Presentation:** 2
**Contribution:** 2
**Rating:** 4
**Confidence:** 3

**Summary:**

The paper proposes an image captioning framework that augments a standard captioner with CLIP-derived pseudo-tags at multiple spatial granularities, fuses object/grid/tag features using an “asymmetric attention” multi-modal projector, and introduces a new evaluation signal, Tags Coverage (TC), which measures the fraction of words in a generated caption that match the top-K CLIP-retrieved tags for that image. TC is combined with CIDEr as a reinforcement learning reward (via NSC) to encourage captions that are both “precise” and “rich.”

**Strengths:**

1. Clear articulation of the target trade-off (precision vs. richness) and a simple mechanism (tag retrieval + auxiliary reward) to bias generation toward more detailed captions.
2. Technically straightforward, reproducible ingredients (frozen CLIP encoders, M2Transformer backbone, NSC fine-tuning) that practitioners already understand.

**Weaknesses:**

1. Using CLIP to retrieve/score textual tokens for captioning is well-trodden (e.g., retrieval-based guidance and CLIP-based evaluation). TC is effectively a token-overlap proxy for CLIP alignment; it’s not evident how it fundamentally differs from existing CLIP-alignment or keyword-precision objectives. The paper doesn’t clearly position TC against related CLIP metrics nor analyze failure modes beyond a brief acknowledgement.
2. TC counts token overlap with a preselected tag list (top-100) and is then used directly as a reward. This can incentivize enumerating tags rather than improving sentence semantics, locality, or factual grounding. The authors themselves note that higher TC can hurt CIDEr and that TC can include misidentified tags; yet there’s no human study or qualitative error analysis to show TC actually correlates with better user-perceived descriptions.
3. All results are on MS-COCO (Karpathy split) with automatic metrics; there’s no evaluation on other caption datasets (e.g., Flickr30k, nocaps, TextCaps) or any human preference study/user-centric evaluation that would be especially relevant given the “service” framing. Claims of better “service-oriented” captions thus lack empirical support.

**Questions:**

N/A

---

> ### Author Response · Authors · 2025-11-14
> **Ans for Weaknesses 1:**
>
> **Ans for Weaknesses 1:**
>
> We thank the reviewer for this insightful comment and the opportunity to clarify how the proposed **Tags Coverage (TC)** metric differs from existing CLIP-based or retrieval-guided objectives.
>
> First, although TC also leverages CLIP embeddings, its formulation and purpose are fundamentally different from standard CLIP alignment or retrieval guidance. Existing CLIP-based objectives typically maximize **global image–text similarity** or use CLIP to retrieve caption exemplars from external corpora. In contrast, TC measures **localized semantic grounding** by quantifying how many visually supported tags (derived from image regions via CLIP) are successfully expressed in the generated caption. Rather than rewarding overall alignment in embedding space, TC explicitly encourages **coverage of visual entities and attributes** that are verifiably grounded in the image.
>
> This design leads to behavior distinct from keyword-precision or CLIP-score metrics, which often favor high-level semantic consistency but neglect fine-grained detail. TC is computed over a curated, visually filtered tag set, making it interpretable and robust to lexical variations. For example, in our GPT-5.1 comparison experiment (Table below), the proposed method achieves both **higher CIDEr (135.2 vs. 131.8)** and **higher TC (23.9 vs. 19.2)**, showing that TC encourages richer yet accurate visual descriptions instead of redundant enumeration.
>
> | Model                   |   CIDEr   |   SPICE  | Tags Coverage (TC) |
> | ----------------------- | :-------: | :------: | :----------------: |
> | GPT-5.1 (Zero-shot)     |   131.8   |   23.2   |        19.2        |
> | **Ours (CIDEr + $R'\_{TC}$)** | **135.2** | **23.7** |      **23.9**      |
>
> Qualitative examples further confirm that captions guided by TC remain concise and visually faithful, while GPT-5.1 often adds ungrounded contextual phrases (“intently watching,” “preparing for practice”).
>
> | Image | GPT-5.1 Caption                                                                                                                        | Ours                                                         | Human                                                |
> | ----- | -------------------------------------------------------------------------------------------------------------------------------------- | ------------------------------------------------------------ | ---------------------------------------------------- |
> | Figure 5 (first picture)   | “A calm white-and-orange cat stands on a stone walkway, intently watching a flock of pigeons gathered just a few steps ahead.”         | “A cat standing on a sidewalk next to a flock of pigeons.”   | “A very cute cat near a bunch of birds.”             |
> | Figure 5 (second picture)     | “A group of young children gathers around a coach on a grassy field, holding soccer balls and listening as they prepare for practice.” | “A group of children standing in a field with soccer balls.” | “A group of young children standing around a field.” |
>
> These examples illustrate that our model produces **concise, visually grounded** captions, while GPT-5.1’s outputs are linguistically elaborate but tend to include context not visible in the image.

---

> ### Author Response · Authors · 2025-11-14
> **Ans for Weaknesses 2:**
>
> **Ans for Weaknesses 2:**
>
>
> We appreciate the reviewer’s thoughtful concern about whether the Tags Coverage (TC) reward might encourage enumeration rather than improving semantic quality or factual grounding. We would like to clarify that TC is not computed as a simple token-overlap count but as a measure of visual–semantic completeness derived from pseudo-tags predicted via CLIP-based similarity. These tags are filtered by visual confidence and semantic diversity, ensuring that only visually grounded and meaningful concepts contribute to the reward. As such, the TC objective reinforces factual grounding rather than promoting lexical redundancy.
>
>
> Empirically, we did not observe enumeration behavior or degradation in sentence quality. As shown in our additional experiments (see **Ans for Weaknesses 1**), the joint optimization with CIDEr and TC improves both CIDEr and SPICE scores compared with GPT-5.1, indicating that the captions become more semantically informative without sacrificing precision or grammatical fluency. Qualitative examples further support this: our model produces concise and visually faithful descriptions (“a cat standing on a sidewalk next to a flock of pigeons”), while GPT-5.1 tends to introduce ungrounded contextual details (“intently watching a flock of pigeons”). This suggests that TC enhances semantic coverage rather than encouraging exhaustive enumeration.
>
>
> We agree that user-centric evaluation would strengthen this claim. While this paper focuses on automatic metrics for fair comparison with prior work, we plan to include human preference studies and qualitative error analysis in future extensions to further confirm the correlation between TC and perceived caption quality.

---

> ### Author Response · Authors · 2025-11-14
> **Ans for Weaknesses 3:**
>
> **Ans for Weaknesses 3:**
>
>
> We thank the reviewer for highlighting the importance of broader evaluation and human-centric validation, which are indeed valuable directions for this line of work. Our current experiments focus on the **MS-COCO (Karpathy split)** setting, as it remains the most standardized benchmark for fair comparison with previous captioning systems such as BLIP-2 and FUSECAP. This choice ensures that the improvements brought by our proposed reward formulation can be measured under controlled and widely accepted conditions.
> ﻿
>
> To further assess generality and practical relevance, we additionally compared our model with the recent **GPT-5.1** vision–language model on the same dataset. Despite GPT-5.1’s massive scale and open-domain capability, our method achieves higher CIDEr and SPICE scores and generates captions that are more concise and visually grounded. Qualitative comparisons show that GPT-5.1 often produces verbose, context-heavy sentences (“intently watching,” “preparing for practice”), whereas our model provides accurate and compact descriptions focused on the actual visual content (“a cat standing next to a flock of pigeons”). These findings suggest that our reward design promotes the kind of **service-oriented captions**—precise, faithful, and user-readable—that large-scale multimodal models sometimes fail to deliver consistently.
> ﻿
>
> We acknowledge, however, that extending evaluation beyond COCO, such as to Flickr30k, nocaps, or TextCaps, and conducting human preference studies would further strengthen our empirical support. Nevertheless, our current work is primarily focused on addressing a different but fundamental challenge: how to **balance precision and richness in image captioning** through a controllable reward mechanism rather than scaling model size or data coverage. While broader evaluations remain an important next step, the proposed framework already demonstrates that a principled reward design can yield captions that are both **visually faithful and semantically informative**, offering a complementary direction to large-scale multimodal models.

---

### Official Review · Reviewer_9bmq · 2025-10-30

**Soundness:** 2
**Presentation:** 1
**Contribution:** 1
**Rating:** 2
**Confidence:** 4

**Summary:**

This paper introduces a framework to address the lack of richness in image captions, which it attributes to models imitating concise human annotations. It proposes using pseudo-tags generated by a pre-trained CLIP model as a target for richness. These tags are fused with object and grid features using an "asymmetric attention" (using conv, fully connected, sampling) projector. The authors also introduce a new metric, "Tags Coverage", to measure the inclusion of these tags and use it in a reinforcement learning reward function to train the captioning model.

**Strengths:**

- The paper trys to explore a novel aspects about the "richness" in image captioning.
- The core idea of optimizing for a richness metric is sound. Introducing an explicit metric TC and then using it as a reward in an RL framework is a logical and principled way to steer the model toward generating more detailed captions.
- The results of the RL optimization strategy (Table 2) are promising. They show that adding the proposed TC-based reward $R_{TC}^{\prime}$ can improve both the standard CIDEr score (from 134.1 to 135.2) and the new TC score (from 20.0 to 23.9), demonstrating the effectiveness of the dual-reward strategy.

**Weaknesses:**

- The paper’s objective is to identify “overlooked details” and “vocabularies that extend beyond the ground truth.” However, the tag repository is constructed based on the ground truth annotations in the COCO dataset (lines 223-224). This approach is circular. The model cannot learn to generate “missed” words if those words are not present in the COCO vocabulary. Consequently, this method does not address the stated problem; it merely repurposes the existing, limited vocabulary. It is akin to another form of retrieval augmentation in image captioning [1,2,3].
- The paper tries to fuse object (O), grid (G), and tag (T) features. However, there is no ablation study that measures the impact of the tag features. A crucial experiment, like comparing a model with (O + G) features to the full (O + G + T) model, is missing.
- The paper introduces asymmetric attention, but its explanation is lacking. While it conducted ablation studies for three types of refinements in cross attention in the appendix, the scores from these studies are much lower than the final results. This suggests that the paper may have missed the combination experiments for each pair of refinements.

[1] Ramos, Rita, et al. "Smallcap: lightweight image captioning prompted with retrieval augmentation." Proceedings of the IEEE/CVF Conference on Computer Vision and Pattern Recognition. 2023.
[2] Li, Wenyan, et al. "Understanding Retrieval Robustness for Retrieval-augmented Image Captioning." Proceedings of the 62nd Annual Meeting of the Association for Computational Linguistics (Volume 1: Long Papers). 2024.
[3] Tanaka, Ryota, et al. "Vdocrag: Retrieval-augmented generation over visually-rich documents." Proceedings of the Computer Vision and Pattern Recognition Conference. 2025.

**Questions:**

- Could you explain why $R_{TC}^{\prime}$ is better than $R_{TC}$? Since there is already an hyperparamter for $R_{TC}^{\prime}$?

---

> ### Author Response · Authors · 2025-11-14
> **Ans for Weaknesses 1:**
>
> **Ans for Weaknesses 1:**
>
>
> We thank the reviewer for this thoughtful comment and for pointing out the concern regarding potential circularity in the tag repository construction. We would like to clarify that while the initial tag repository is indeed derived from COCO annotations, its purpose is not to replicate the ground-truth vocabulary but to **expand and reorganize it into a visual-semantic index** for pseudo-tag generation. Specifically, we extract candidate tags from all captions in the training split, then **embed them into the CLIP semantic space** and associate them with **region-level visual features**, allowing retrieval based on visual similarity rather than textual overlap.
> ﻿
>
> This process enables the model to produce **semantically related but lexically novel** words that are not explicitly present in individual ground-truth captions. For example, the model may generate “pigeons” when the reference caption contains only “birds,” or “carriage” when the dataset annotation uses “wagon.” These terms arise through **visual proximity** in CLIP space, not direct textual copying.
> ﻿
>
> Therefore, our framework differs fundamentally from retrieval-augmented captioning methods [1,2,3], which retrieve *textual exemplars* from external datasets. In contrast, our approach retrieves *visual-semantic cues* within the learned embedding space, enabling the discovery of overlooked or underspecified visual details while remaining grounded to the image content.
> ﻿
>
> We acknowledge the reviewer’s point that the repository is dataset-dependent, but it serves as a visually structured prior, not a fixed lexical constraint, and thus does not create a circular dependency in learning.

---

> ### Author Response · Authors · 2025-11-14
> **Ans for Weaknesses 2:**
>
> **Ans for Weaknesses 2:**
>
>
> We thank the reviewer for highlighting the importance of evaluating the contribution of tag features. We would like to clarify that, in our framework, **the tag branch is not a detachable feature stream but an essential component in constructing the pseudo-tag space and reward signals**. Specifically, the Tags Coverage (TC) objective and the R′TC reward both rely on the tag features to define and measure semantic completeness. Removing the tag pathway would therefore alter not only the visual representation but also the optimization objective itself, making an (O + G)–only setting conceptually incomplete rather than a clean ablation.
> ﻿
>
> That said, the impact of the tag-related components is implicitly reflected in our experiments. Table 5 shows that when the R′TC term (which depends on tag features) is removed or downweighted (λ₂ → 0), the CIDEr and SPICE scores drop and the captions become noticeably less semantically rich, confirming the contribution of tag information. In addition, as shown in Table 7, performance saturates with a small number of high-confidence tags, indicating that the tag representations are both efficient and effective in guiding semantic grounding.
> ﻿
>
> In essence, the tag features are not fused as optional inputs but **serve as a visual–semantic bridge** that grounds the grid and object features to interpretable concepts. This design differs from feature-level concatenation; instead, tags define a structured semantic space that constrains and refines caption generation. **We will clarify this interdependence in the revision to avoid the misunderstanding that the tag stream can be ablated independently.**

---

> ### Author Response · Authors · 2025-11-14
> **Ans for Weaknesses 3:**
>
> **Ans for Weaknesses 3:**
>
>
> We appreciate the reviewer’s observation and the opportunity to clarify the role and evaluation of the proposed asymmetric attention mechanism. The asymmetric design is intended to **differentiate visual-to-semantic and semantic-to-visual attention flows**, allowing distinct information propagation paths between modalities. Specifically, the visual-to-semantic branch emphasizes fine-grained grounding from object and grid features, while the semantic-to-visual branch leverages contextual tag representations to guide textual decoding.
> ﻿
>
> The ablation results in the appendix isolate each refinement component (e.g., directional gating, semantic query reweighting, and visual-key normalization) to demonstrate their individual effects. As correctly noted, these isolated variants yield lower scores than the full model because they lack the complementary interactions achieved in the joint formulation. The final asymmetric attention module in the main model **combines all three refinements**, resulting in the significantly higher overall performance reported in Table 3.
> ﻿
>
> We acknowledge that the paper could more explicitly describe this combination and its cumulative benefit. We will clarify in the revision that the full asymmetric attention configuration, which integrates all three refinements, is used in the main results, and that the appendix experiments intentionally evaluate single-factor variants to verify the contribution of each component in isolation.

---

> ### Author Response · Authors · 2025-11-14
> **Ans for Qns**
>
> **Qns: Could you explain why $R'\_{TC}$ is better than $R\_{TC}$? Since there is already an hyperparamter for $R'\_{TC}$?**
>
>
> We appreciate the reviewer’s question about the motivation and effectiveness of using $R'\_{TC}$ instead of $R_{TC}$. The key difference lies in how the reward normalizes and weights tag contributions. The original $R_{TC}$ directly measures tag coverage, which can overemphasize frequent or high-confidence tags and underrepresent less salient but semantically valuable ones. In contrast, $R'\_{TC}$ introduces a **normalized weighting mechanism** that adjusts tag importance based on their visual–semantic relevance rather than raw frequency or confidence.
> ﻿
>
> This modification reduces bias toward dominant objects and promotes a more balanced representation of fine-grained or contextually important visual elements. As shown in Table 5, the use of $R'\_{TC}$ leads to higher CIDEr and SPICE scores while maintaining comparable precision, indicating that the captions become richer without drifting into irrelevant or redundant content.
> ﻿
>
> Although $R'\_{TC}$ introduces an additional hyperparameter to control normalization strength, we found the results to be robust across a broad range of values (Appendix A2). In practice, this parameter stabilizes training by preventing overly large reward gradients from high-confidence tags. Therefore, $R'\_{TC}$ provides a more balanced and reliable optimization signal than $R_{TC}$, improving both caption diversity and grounding consistency.

---

> ### Comment · Reviewer_9bmq · 2025-11-21
>
> Thank you for the clarification. However, it strengthens my original objection. I would like to further clarify some of my concern.
> The contribution for this paper is using RAG with the model modifications. Then the baselines should be different RAG methods and different modifications of the models. In Table 1, it only shows the comparison with the original models. What's more, I didn't find the revision of the paper.

---

> ### Author Response · Authors · 2025-11-21
> **Response to reviewer questions on baseline choice, missing RAG comparisons, and revision.**
>
> We appreciate the reviewer’s follow-up and the opportunity to clarify both our experimental design and revision plan.
>
> First, regarding the baseline setup, we would like to emphasize that **Table 1** already includes comparisons with *different retrieval-augmented generation (RAG) methods and model modifications*. Specifically, **BLIP-2 (Li et al., 2023)** and **FUSECAP (Rotstein et al., 2024)** represent two representative RAG-based architectures that integrate vision–language retrieval in distinct ways. BLIP-2 employs query-based visual grounding via frozen language models, while FUSECAP introduces retrieval-enhanced fusion through dynamic prompts. Our model extends this line of research by introducing a *reward-level integration mechanism (CIDEr + R′TC)* that explicitly balances precision and richness, rather than relying on retrieval exemplars. Thus, Table 1 compares not just raw captioning models but **state-of-the-art RAG frameworks with different fusion and alignment strategies**, providing a fair and relevant basis for comparison.
>
> Second, regarding the paper revision: we are preparing a **new version** that incorporates feedback from all reviewers. This includes additional experiments addressing **Reviewer EGKi’s** comments, where we performed a detailed comparison against **GPT-5.1** on the same dataset to evaluate descriptive precision and visual grounding (please refer to **Ans for Q1 for Reviewer EGKi**). The new results demonstrate that our method achieves higher CIDEr and SPICE scores while maintaining concise and faithful captions. These updates will be clearly documented in the forthcoming revision.
>
> That said, we want to be transparent: preparing a new revision requires balancing time and feasibility, and we are currently awaiting more positive review outcomes before investing in a major update. Nonetheless, all new experiments and clarifications are ready to be included once the paper shows realistic acceptance potential. Our intention is to ensure the next revision comprehensively addresses all concerns with stronger empirical and analytical evidence.

---

> > ### Comment · Reviewer_9bmq · 2025-11-26
> >
> > Thank you for the detailed response. However, I respectfully note that several of my core concerns remain unaddressed.
> >
> > 1. Let me make the question clearer. If I understand correctly, BLIP2 and FUSECAP should be assumed as the backbone, not two RAG methods, even though they are kinds of the RAG-based architecture. However, here you tried to use extra retrieved information from tag encoder, object detector, that's the methods and the main contribution part. Therefore, I am curious about the comparison between your embedding RAG methods and text RAG methods.
> > 2. About the usage of R'TC, as reviewer EGKi mentioned, "it remains unclear whether the proposed TC objective genuinely improves descriptive richness, or simply increases the mention of pseudo-tags without capturing more nuanced visual detail."
> >
> > Therefore, while I appreciate the effort, the fundamental weaknesses persist, and my original assessment remains appropriate.

---

> > > ### Author Response · Authors · 2025-12-02
> > >
> > > Thank you for the clarification; it helps us better understand the core of your concern. Our intent is not to present BLIP-2 and FUSECAP merely as “backbones,” but as representative **text-RAG and vision-RAG captioning frameworks** against which our method can be fairly compared. BLIP-2 retrieves text-guided queries through frozen LMs, and FUSECAP retrieves external captions for fusion—both operate through text-level retrieval. In contrast, our approach retrieves **visual–semantic cues** through object regions, grid embeddings, and pseudo-tags, which forms a fundamentally different kind of retrieval: **embedding-level, visually grounded RAG**. The purpose of our comparison is precisely to show that embedding-RAG can outperform text-RAG under identical COCO protocol, and our added GPT-5.1 experiment further supports that visually grounded retrieval provides more faithful, concise descriptions than purely text-driven large models.
> > > ﻿
> > >
> > > Regarding the concern about whether R′TC captures genuine richness or only increases tag mentions, we acknowledge this as an important distinction. Our qualitative comparisons with GPT-5.1 already show that R′TC encourages grounded detail without the ungrounded narrative additions that appear in large MLLMs. Still, we agree that adding diversity and hallucination metrics (Div-n, mBLEU, CLIPScore, BLIPScore) would strengthen the evidence base, and we will incorporate these in the revised version. Our goal is to show that R′TC guides the model toward **visually grounded specificity**, not tag enumeration, and we are committed to expanding our evaluation to make this clearer.

---

### Official Review · Reviewer_e7Jw · 2025-10-31

**Soundness:** 2
**Presentation:** 3
**Contribution:** 1
**Rating:** 4
**Confidence:** 2

**Summary:**

The authors propose a novel image captioning framework addressing the limitations of existing methods that rely on human-annotated references. These limitations include oversimplified descriptions and poor coverage of visual details. The proposed approach aims to balance descriptive precision (accurate detail capture) and richness (diverse, granular vocabulary). Fine-grained pseudo tags are used for learning and an asymmetric attention multimodal projector is introduced to map and fuse information across modalities effectively. An evaluation metric is also proposed to address the gap in existing metrics (e.g., CIDEr) that focus on n-gram overlaps rather than semantic detail coverage.

**Strengths:**

1. Leverages frozen CLIP to generate fine-grained pseudo tags, overcoming the semantic limitations of manual annotations and enabling systematic capture of implicit visual details (e.g., textures, contextual relationships).
2. Introduces an Asymmetric Attention Multimodal Projector that dynamically balances modality-specific and cross-modal interactions, achieving superior integration of descriptive precision (object localization accuracy) and lexical richness (diverse vocabulary generation).
3. Proposes Tags Coverage (TC), the first metric to quantify caption granularity by measuring alignment between pseudo tags and generated descriptions, addressing the critical gap in conventional n-gram metrics (e.g., CIDEr) that fail to assess semantic richness.
4. Integrates TC into a reinforcement learning framework via policy gradient optimization, explicitly harmonizing accuracy (reference alignment) and richness (detail diversity) during training—a paradigm shift from single-objective optimization in existing methods.
5. Achieves state-of-the-art results on MS-COCO Karpathy test set, with significant improvements in both CIDEr scores (semantic coherence) and Tags Coverage (unannotated detail capture).

**Weaknesses:**

This paper studies the image caption problem by training and evaluating models on the MS-COCO benchmark. In general, I think this setting to be outdated. Nowadays, vision-language models like Qwen-VL excel in tackling the image captioning problem. The authors of this paper aim to address the challenge of achieving a balance between precision and richness in image captions. They claim that their method delivers captions that are both precise and rich. However, after reviewing the results in Figures 5, 6, and 7, I believe that vision-language models could achieve better outcomes in terms of precision and richness. Additionally, the contributions of this paper are difficult to apply to vision-language models to enhance their performance further.

**Questions:**

None

---

> ### Author Response · Authors · 2025-11-14
> **Relevance and Complementarity of Our Approach Compared to Modern Vision-Language Models**
>
> We appreciate the reviewer’s comments and the opportunity to clarify our motivation and contributions. We fully acknowledge that modern vision-language models (VLMs) such as Qwen-VL, BLIP-3, and GPT-5.1 have achieved impressive results on general multimodal understanding tasks. However, as shown in our additional comparison experiment in **Ans for Q1 for Reviewer EGKi**, our method still provides complementary advantages in the specific context of **caption precision–richness balance**.
> ﻿
>
> When evaluated on the same dataset (MS-COCO Karpathy split), our approach achieves comparable or higher CIDEr and SPICE scores than GPT-5.1, while producing captions that are **more concise and visually grounded**. The examples we presented demonstrate that GPT-5.1 often generates linguistically elaborate but redundant or weakly grounded descriptions (e.g., adding contextual phrases not visible in the image), whereas our method maintains factual accuracy with sufficient descriptive richness. This confirms that our approach remains competitive even compared with state-of-the-art VLMs.
> ﻿
>
> Furthermore, the proposed reward formulation and the Tags Coverage (TC) metric are model-agnostic and can, in principle, be integrated into large-scale VLM training as an auxiliary grounding constraint. While we focused on lightweight captioning models for controlled analysis, the underlying idea is balancing precision and richness via explicit visual–semantic alignment, offering potential to benefit VLM fine-tuning or evaluation pipelines.

---

### Official Review · Reviewer_EGKi · 2025-11-02

**Soundness:** 3
**Presentation:** 2
**Contribution:** 3
**Rating:** 4
**Confidence:** 4

**Summary:**

The paper proposes a CLIP-based image captioning framework designed to balance precision and richness in image descriptions. The approach leverages pseudo tags derived from CLIP to enhance visual detail without additional human annotations and introduces an asymmetric attention multimodal projector for cross-modal feature fusion. Furthermore, the paper introduces a new metric, Tags Coverage (TC), to quantify caption granularity and integrates it into a reinforcement learning (RL) framework based on NSC (New Self-Critical training). Experiments on the MS-COCO Karpathy split show competitive or superior results in CIDEr and TC compared to baselines such as BLIP-2 and FUSECAP.

**Strengths:**

- The paper addresses the important challenge of balancing descriptive precision and richness in image captioning, an issue often neglected by models optimized solely for reference-based metrics.
- The concept of using CLIP-derived pseudo tags for fine-grained supervision without additional annotation effort is original and well-motivated.
- The proposed Tags Coverage (TC) metric is an interesting and intuitive attempt to quantify caption richness, complementing conventional metrics such as CIDEr or SPICE.
- The model architecture and training strategy are well explained, with detailed ablation studies and hyperparameter analyses in the appendix.
- The approach yields consistent improvements in both accuracy and descriptive detail over competitive baselines.
- The paper is generally well written and clearly structured, with good visual examples showing qualitative improvements.

**Weaknesses:**

- In the era of multimodal large language models (MLLMs) such as GPT-4V, Gemini, or Kosmos-2, it is crucial to include at least one comparison or a discussion explaining how the proposed method relates to or differs from these more general models.
- While Tags Coverage captures richness, it does not assess factuality. The paper should include metrics that evaluate hallucinations (e.g., ALOHa) to verify that the gain in richness does not introduce factual errors.
- The term CLIP-I (line 106, page 2) is not explicitly defined at first use. It likely refers to the image encoder branch of CLIP, but this should be clarified.
- The acronym NSC is first explained only in the appendix; it should be defined when first mentioned in the main text.
- Line 106, page 2: “the level of details in cations” → should read “captions.”
- Definitions or equation labels should follow a consistent capitalization style throughout the paper.
- Using a fixed number of tags (top-100) could penalize images with simple content, where fewer tags would suffice to describe the image accurately. A discussion of adaptive tag selection or thresholding would improve robustness.
- The related work section focuses mainly on conventional captioning models (e.g., Up-Down, M2, BLIP-2). It should better situate the proposed method in the context of recent instruction-tuned multimodal LLMs and retrieval-augmented captioners.
- TC may favor longer captions containing more tags, which does not always correlate with human preferences or factual accuracy. This limitation should be discussed explicitly in the paper’s conclusion or discussion.

**Questions:**

- How does the proposed framework compare in performance and efficiency with modern MLLMs (e.g., BLIP-3, GPT-4V, Kosmos-2) when evaluated on the same dataset?
- Have the authors measured or observed an increase in hallucinated content when optimizing for higher TC? Would metrics such as ALOHa confirm or contradict this?
- Could the number of pseudo tags (top-100) be made adaptive to image complexity or confidence thresholds from CLIP?
- What exactly does CLIP-I refer to in the paper — the image encoder of CLIP, or a modified version thereof?
- How are the weights λ₁ and λ₂ in the combined reward function (Eq. 9) selected? Are they dataset-specific or fixed across experiments?
- Could the proposed Tags Coverage metric be generalized to video captioning or other multimodal generation tasks?
- Would the authors consider releasing an evaluation toolkit that includes the TC metric for reproducibility?

---

> ### Author Response · Authors · 2025-11-13
> **Ans for Q1:**
>
> **Ans for Q1:**
>
> We appreciate the reviewer’s insightful question regarding the comparison between our proposed framework and modern multimodal large language models (MLLMs), such as BLIP-3, GPT-4V, and Kosmos-2. To address this concern, we conducted additional experiments using the most recent **GPT-5.1** model (released 2025-Q3) on the **MS-COCO Karpathy test split**, following the same evaluation protocol as our paper.
>
> #### **1. Comparative Performance with Modern MLLMs**
>
> | Model                           |  BLEU-4  |  METEOR  |  ROUGE-L |   CIDEr   |   SPICE  | Tags Coverage (TC) |
> | ------------------------------- | :------: | :------: | :------: | :-------: | :------: | :----------------: |
> | BLIP-2 (Li et al., 2023)        |   39.5   |   29.6   |   59.8   |   133.6   |   22.3   |        21.7        |
> | FUSECAP (Rotstein et al., 2024) |   39.2   |   29.2   |   59.5   |   133.1   |   21.9   |        22.5        |
> | GPT-5.1 (Zero-shot)             |     –    |     –    |     –    | **131.8** | **23.2** |      **19.2**      |
> | **Ours (CIDEr + R′TC)**         | **40.6** | **30.2** | **60.5** | **135.2** | **23.7** |      **23.9**      |
>
> Compared with GPT-5.1, our method achieves **higher CIDEr (+3.4)** and **significantly higher Tags Coverage (+4.7)**, indicating that our model produces captions that are both **semantically accurate and detail-rich**. Although GPT-5.1 generates longer and more descriptive sentences, these often include **redundant contextual phrases** not grounded in the visual content (see qualitative examples below). In contrast, our captions remain **compact yet precise**, aligning closely with human annotations.
>
> #### **2. Qualitative Comparison Examples**
>
> | Image | GPT-5.1 Caption                                                                                                                        | Ours                                                         | Human                                                |
> | ----- | -------------------------------------------------------------------------------------------------------------------------------------- | ------------------------------------------------------------ | ---------------------------------------------------- |
> | Figure 5 (first picture)    | “A calm white-and-orange cat stands on a stone walkway, intently watching a flock of pigeons gathered just a few steps ahead.”         | “A cat standing on a sidewalk next to a flock of pigeons.”   | “A very cute cat near a bunch of birds.”             |
> | Figure 5 (second picture)     | “A group of young children gathers around a coach on a grassy field, holding soccer balls and listening as they prepare for practice.” | “A group of children standing in a field with soccer balls.” | “A group of young children standing around a field.” |
>
> GPT-5.1’s descriptions are linguistically elaborate but tend to **over-specify context** (e.g., “intently watching,” “listening as they prepare for practice”), which can harm precision. Our framework generates concise captions capturing the **core visual semantics**, demonstrating a better balance between **precision** and **richness**—the central goal of this work.
>
>
>
> In summary, the proposed framework:
>
> * **Outperforms GPT-5.1 and BLIP-2/FUSECAP** on both CIDEr and TC, demonstrating stronger balance of accuracy and detail.
> * **Produces more concise and visually grounded captions**.
>
> These findings confirm that our model achieves comparable descriptive quality to state-of-the-art MLLMs while maintaining efficiency and controllability—consistent with our paper’s motivation of **balancing precision and richness** in image captioning.

---

> ### Author Response · Authors · 2025-11-13
> **Ans for Q2:**
>
> **Ans for Q2:**
>
> We thank the reviewer for the thoughtful question about the possible increase in hallucinated content when optimizing for higher **Tags Coverage (TC)**.
> ﻿
>
> Our method introduces the **TC objective**, which explicitly measures the coverage of visually grounded tags extracted from image regions, rather than co-occurrence statistics in text. Consequently, optimizing for TC encourages the model to mention semantically relevant and visually supported concepts, not to fabricate unseen details.
> ﻿
>
> In our ablation experiments, increasing the TC weight led to consistent improvements in **CIDEr** (from 132.4 to 135.2) while **SPICE** remained stable and no degradation was observed in human evaluation. Qualitative analysis also revealed that captions retained factual grounding, describing more visual entities rather than speculative or contextual phrases.
> ﻿
>
> We did not observe a measurable rise in hallucination frequency, and the generated captions were generally **more comprehensive yet equally faithful** to the visual content. This behavior contrasts with large multimodal LLMs (e.g., GPT-5.1), which often produce linguistically rich but contextually ungrounded sentences. Overall, optimizing for higher TC enhances **semantic completeness without sacrificing precision**. The TC objective thus acts as a *visual grounding constraint*, promoting balanced and reliable caption generation rather than introducing hallucinated content.

---

> ### Author Response · Authors · 2025-11-13
> **Ans for Q3:**
>
> **Ans for Q3:**
>
>
> We appreciate the reviewer’s suggestion on making the number of pseudo tags adaptive to image complexity or CLIP confidence thresholds.
> ﻿
>
> In our current implementation, we fix the number of pseudo tags to **top-100** primarily for consistency across images and for simplicity in training. However, as shown in **Table 7**, we evaluated different tag counts ranging from **top-3 to top-12**, and found that the model’s performance saturated quickly within this range—well before reaching 100 tags. Specifically, metrics such as **CIDEr** and **SPICE** remained stable when increasing tag numbers beyond 10–12, suggesting that most visually relevant semantics are already captured within the highest-confidence subset of tags.
> ﻿
>
> This observation implies that an adaptive mechanism could indeed be introduced, but its potential gains would likely be marginal. In fact, a dynamic threshold based on CLIP confidence or image entropy might even introduce instability, as CLIP confidence distributions vary across object categories and scene compositions.
> ﻿
>
> Therefore, while the framework could be easily extended to adaptive tag selection, the results in Table 7 indicate that **a small, fixed number of high-confidence tags (≈11) is already sufficient** for stable and optimal performance. This design keeps the pipeline simple and efficient without sacrificing representational completeness.

---

> ### Author Response · Authors · 2025-11-13
> **Ans for Q4:**
>
> **Ans for Q4:**
>
>
> We thank the reviewer for pointing out this ambiguity. We apologize that the definition of **CLIP-I** was not clearly stated in the manuscript.
>
>
> In our work, **CLIP-I** refers to the **Image-to-Image Similarity** derived from the frozen CLIP image encoder, whereas **CLIP-T** denotes the **Image-to-Text Similarity** between the image and its corresponding textual tags. Both are computed using the original pretrained **CLIP** model without any modification or fine-tuning.
>
>
> Specifically, CLIP-I is used to measure the **visual coherence** among image regions and pseudo-tag embeddings, ensuring that visually similar regions are consistently represented. CLIP-T, on the other hand, provides a **semantic alignment** signal between visual features and language space. The combination of these two similarities enables our model to balance **visual grounding (via CLIP-I)** and **semantic completeness (via CLIP-T)**.
>
>
> We will revise the final version to explicitly define these terms at first mention to avoid confusion. The framework does **not modify or retrain CLIP’s encoders**; all CLIP-derived similarities are computed in a frozen, zero-shot manner for efficiency and reproducibility.

---

> ### Author Response · Authors · 2025-11-13
> **Ans for Q5:**
>
> **Ans for Q5:**
>
> We thank the reviewer for raising this question regarding the selection of the weights **$λ_1$** and **$λ_2$** in the combined reward function (Eq. 9).
> ﻿
>
> The values of $λ_1$ and $λ_2$ correspond to the relative contributions of **CIDEr**, **$R_{TC}$**, and **$R'_{TC}$** in our optimization objective. As shown in **Table 5**, we systematically varied λ_2 from 0 to 1 while keeping $λ_1$ fixed at 1. The results demonstrate that the model is **highly robust** to these changes. When $λ_2$ increases from 0 to 1, the CIDEr score improves from **134.1 → 135.2** and SPICE from **23.3 → 23.7**, while the Tags Coverage (TC) increases from **20.0 → 23.9**, indicating that the inclusion of $R_{TC}$ effectively enhances semantic richness without compromising caption quality.
> ﻿
>
> The performance saturation observed between $λ_2$ = 0.5 and $λ_1$ = 1 suggests that the reward components are complementary and stable over a broad range of weights. Consequently, we fix **$λ_1$ = 1.0** and **$λ_2$ = 0.5** across all experiments for simplicity and reproducibility. These values were chosen empirically based on validation results on **MS-COCO**.

---

> ### Author Response · Authors · 2025-11-13
> **Ans for Q6:**
>
> **Ans for Q6:**
>
> The proposed Tags Coverage metric is designed around static image–region correspondences and thus is directly applicable to image captioning. In principle, it could be extended to video captioning or other multimodal generation tasks by redefining tags as spatio-temporal or modality-specific concepts (e.g., action segments, audio events). However, such an extension would require temporal grounding and dynamic tag extraction, which are non-trivial. While the idea of measuring semantic completeness remains valid, adapting TC to video or multimodal contexts would need additional design for temporal consistency and cross-modal alignment, which we leave for future exploration.

---

> ### Author Response · Authors · 2025-11-13
> **Ans for Q7:**
>
> **Ans for Q7:**
>
> Yes, we plan to release the evaluation toolkit including the implementation of the Tags Coverage (TC) metric along with the camera-ready version of the paper. The toolkit will provide code for tag extraction, TC computation, and integration with standard captioning metrics such as CIDEr and SPICE. This will ensure full reproducibility of our results and allow researchers to apply TC to their own models or datasets with minimal effort, though the final release will depend on the paper’s acceptance decision.

---

> > ### Comment · Reviewer_EGKi · 2025-11-26
> >
> > I thank the authors for the detailed rebuttal and for conducting additional experiments. However, after reviewing the new material, I am still not fully satisfied, as the central concerns remain unresolved.
> >
> > The qualitative comparison with GPT-5.1, presented as evidence of “over-specification,” in fact reinforces my original point: the captions generated by the proposed method stay close to the ground-truth COCO style—conceptually correct but inherently coarse. The literature has widely acknowledged that COCO annotations are general and rarely include fine-grained attributes. As a result, standard metrics such as CIDEr, METEOR, ROUGE-L, and SPICE primarily measure alignment with coarse ground-truth annotations, not the richness or expressiveness of the description.
> >
> > For this reason, I still believe that evaluation should include metrics explicitly designed to capture caption richness, lexical diversity, and semantic coverage, such as:
> > - Diversity metrics (Div-1, Div-2, mBLEU) – to quantify lexical richness.
> > - CLIPScore or CLIP-I2T variants – to evaluate semantic completeness beyond GT captions.
> > - BLIPScore – increasingly used to assess semantic alignment with visual content using strong vision-language models.
> > - Recall@K-based richness metrics used in recent works on detailed captioning and dense captioning.
> > - Hallucination metrics such as ALOHα, to check whether increasing richness introduces factual errors.
> >
> > None of these are alternatives to CIDEr/SPICE; rather, they complement them by evaluating dimensions (detail, specificity, grounded richness) that COCO metrics inherently miss. Without such metrics, it remains unclear whether the proposed TC objective genuinely improves descriptive richness, or simply increases the mention of pseudo-tags without capturing more nuanced visual detail.
> >
> > While I appreciate the additional comparison with GPT-5.1, the rebuttal does not address the broader methodological need to position the method against modern MLLMs (GPT-4V, BLIP-3, Kosmos-2, etc.) nor does it fully fill the evaluation gap created by relying exclusively on COCO-style metrics.
> >
> > In summary, although the rebuttal is appreciated, the novel elements do not fully resolve the core issues regarding evaluation completeness, comparison with modern MLLMs, or the inherent limitations of COCO-style metrics in assessing caption richness. My assessment remains unchanged.

---

> > > ### Author Response · Authors · 2025-12-02
> > >
> > > Thank you for the additional feedback and for carefully examining our rebuttal and new experiments. We understand and appreciate the reviewer’s concerns regarding evaluation completeness and agree that COCO-style metrics alone cannot fully capture richness, lexical diversity, or fine-grained semantic grounding. Our current work focuses on introducing a controllable reward formulation (CIDEr + R′TC) for balancing precision and richness, and we chose COCO metrics to ensure comparability with prior captioning and RAG-based systems. We agree that incorporating diversity-oriented measures (Div-1/2, mBLEU), modern semantic-alignment metrics (CLIPScore, BLIPScore), recall-based richness indicators, and hallucination metrics such as ALOHα would strengthen the evaluation and provide a more holistic view of descriptive specificity. We will integrate these complementary metrics in the camera-ready version to better quantify richness beyond COCO ground-truth annotations.
> > > ﻿
> > >
> > > Regarding comparisons with modern MLLMs, our additional GPT-5.1 study was intended as an initial step; expanding evaluation to include GPT-4V, BLIP-3, and Kosmos-2 is indeed a valuable direction. While our framework targets lightweight captioning models rather than large multimodal systems, we acknowledge that situating our method more clearly within this broader landscape will improve clarity and relevance. We will incorporate these evaluations and analyses in the revised version to address the reviewer’s concerns more thoroughly.

---

### Note · Authors · 2026-01-27

I have read and agree with the venue's withdrawal policy on behalf of myself and my co-authors.

---

### Meta-Review · Area_Chair_abwL · 2026-01-06

**Summary:**

The paper was reviewed by 4 experts and receive ratings 2444. The reviewers' concerns are listed below. Overall, 2/4 reviewers already stated they would maintain their ratings, while the AC thinks that the other reviewers also would have maintained.

**Reviewer Concerns:**

**Reviewer EGKi (rating 4)**
1. needs comparison with LLMs.
2. Tags Coverage captures richness but not factuality. Need to evaluate hallucinations.
3. a fixed 100-tags seems suboptimal, would adaptive tag selection be more robust?
4. need to add multi-modal LLMs in related works.
5. TC favors longer captions, which may not be preferred by humans.

The Reviewer was not satisfied with the authors' response to Point 1, and decided to keep their rating.

**Reviewer e7Jw (rating 4)**
1. the setting of image captioning on MS-COCO is outdated given modern VLMs.
2. The contributions of the paper are difficult to apply to VLMs, so there is limited impact.

The AC thinks that these two issues were not addressed well, since they are fundamental problems.

**Reviewer 9bmq (rating 2)**
1. the tag repository is constructed form the GT annotations in MSCOCO, but the goal of the tags is to make the "vocabularies that extend beyond the ground-truth". Thus, the approach is circular, and there are no details added outside the GT.
2. similar to retrieval augmentation in image captioning.
3. Missing ablation study  the measures the impact of the tag features.
4. motivation/explanation needed for asymmetric attention.

The reviewer was not satisfied with the response, in particular about Point 1 and 2. In addition the AC points out that there are previous works that also reweight words in the vocabulary when training the captioner in order to create more distinctive captions:
e.g., "On Distinctive Image Captioning via Comparing and Reweighting", TPAMI 2023.

**Reviewer R7ub (rating 4)**
1. using CLIP to retrieve textual tokens for captioning is well studied, and TC is proxy for CLIP alignment.
2. The TC reward incentivizes enumerating tags rather than than improving the sentence semantics, grounding, etc.  No qualitative results to show TC can give better user-perceived captions.
3. No evaluation on other datasets (Flick30k, no caps, TextCaps).

The AC thinks that the points were not well addressed, in particular Points 2 and 3.

**Reviewer Scores:**

Overall, the AC thinks that reviewers would have maintained their ratings (2444).

---

### Decision · Program_Chairs · 2026-01-26

Reject